# Survey of Dietary Habits and Physical Activity in Japanese Patients with Non-Obese Non-Alcoholic Fatty Liver Disease

**DOI:** 10.3390/nu15122764

**Published:** 2023-06-16

**Authors:** Yoshito Yabe, Kanako Chihara, Natsumi Oshida, Takashi Kamimaki, Naoyuki Hasegawa, Tomonori Isobe, Junichi Shoda

**Affiliations:** 1Medical Sciences, Faculty of Medicine, University of Tsukuba, Tsukuba 305-8575, Japan; yoshitoyabe@outlook.jp (Y.Y.); hanatani8812@gmail.com (K.C.); tiso@md.tsukuba.ac.jp (T.I.); 2Department of Nutrition, Tsukuba Memorial Hospital, Tsukuba 300-2622, Japan; 3Division of Laboratory Medicine, Tsukuba University Hospital, Tsukuba 305-8576, Japan; n.u.al.th1@gmail.com (N.O.); kamimaki0520@yahoo.co.jp (T.K.); 4Division of Gastroenterology, Faculty of Medicine, University of Tsukuba, Tsukuba 305-8575, Japan; hasegawa.naoyuki.gp@u.tsukuba.ac.jp

**Keywords:** non-obese NAFLD, questionnaire survey, dietary habits, physical activity, propensity score

## Abstract

The incidence of non-obese non-alcoholic fatty liver disease (NAFLD), characterized by the presence of a fatty liver in individuals with a normal body mass index, is on the rise globally. Effective management strategies, including lifestyle interventions such as diet and exercise therapy, are urgently needed to address this growing public health concern. The aim of this study was to investigate the association between non-obese NAFLD, dietary habits, and physical activity levels. By elucidating these relationships, this research may contribute to the development of evidence-based recommendations for the management of non-obese NAFLD. The study had a single-center retrospective cross-sectional design and compared clinical data and dietary and physical activity habits between patients with and without non-obese NAFLD. Logistic regression analysis was utilized to investigate the relationship between food intake frequency and the development of NAFLD. Among the 455 patients who visited the clinic during the study period, 169 were selected for analysis, including 74 with non-obese NAFLD and 95 without NAFLD. The non-obese NAFLD group showed a less-frequent consumption of fish and fish products as well as olive oil and canola/rapeseed oil, while they showed more frequent consumption of pastries and cake, snack foods and fried sweets, candy and caramels, salty foods, and pickles compared to the non-NAFLD group. Logistic regression analysis revealed that NAFLD was significantly associated with the consumption of fish, fish products, and pickles at least four times a week. The physical activity level was lower and the exercise frequency was lower in patients with non-obese NAFLD compared to those without NAFLD. The results of this study suggest that a low consumption of fish and fish products and high consumption of pickles may be associated with a higher risk of non-obese NAFLD. Moreover, dietary habits and physical activity status should be taken into consideration for the management of patients with non-obese NAFLD. It is important to develop effective management strategies, such as dietary and exercise interventions, to prevent and treat NAFLD in this patient population.

## 1. Introduction

Non-alcoholic fatty liver disease (NAFLD) is the most prevalent chronic liver disease [1], representing the hepatic manifestation of metabolic syndrome and leading to hepatic steatosis. The etiology of NAFLD includes primary factors, such as insulin resistance, as a consequence of obesity or metabolic syndrome, and secondary factors, which include endocrine disorders, such as hypothyroidism and dyslipidemia, as well as drug-related causes [2]. On the other hand, non-obese NAFLD, also referred to as lean NAFLD, is a condition characterized by the presence of a fatty liver in individuals with a body mass index (BMI) within the normal range [3,4]. According to previous studies, the proportion of individuals with non-obese NAFLD in the general population varies greatly worldwide, with estimates ranging from 5% to 26% [5]. The results of the National Nutrition Survey III and the National Health Survey reported that 10% of the U.S. population had non-obese NAFLD [6]. Studies have suggested that non-obese NAFLD is more prevalent in certain populations, such as in Asian people and women, compared to other groups [7]. In Japan, the prevalence of non-obese NAFLD is predicted to increase to 20.7% by 2040 [8]. It has been reported that regions with a high prevalence of the PNPLA3 gene mutation, which is a disease susceptibility gene for NAFLD, have a higher number of patients with non-obese NAFLD [5]. In particular, the PNPLA3 gene with the GG genotype tends to cause a fatty liver as a result of an accumulation of fat in the liver [9], and a high percentage of Japanese people have this gene [10]. Liver fibrosis, a predictor of prognosis in NAFLD, may be present at an advanced stage upon diagnosis in a significant proportion of individuals with non-obese NAFLD, particularly in females [11]. Furthermore, mortality rates are reported to be higher than those for people with obese NAFLD [12]. Therefore, non-obese NAFLD is likely to become a serious problem in the future, not just in Japan and other Asian nations but also worldwide.

It has been reported that patients with non-obese NAFLD can benefit from lifestyle modifications such as those to diet and exercise, and that the weight loss required to improve non-obese NAFLD is less than that required for obese NAFLD [13]. Previous studies have identified excessive cholesterol and fructose intake, associated weight gain in the normal range, visceral adiposity, and insulin resistance as dietary risk factors for non-obese NAFLD [7]. In the case of general NAFLD, it is believed that a diet high in trans fats, saturated fats, cholesterol, and beverages containing fructose can increase visceral fat, promote the accumulation of lipids in the liver, and stimulate the progression to steatohepatitis and fibrosis [14]. Furthermore, male NASH patients have shown significantly higher weekly intake frequencies of meat, fried foods, Chinese noodles, sweets, and instant foods compared to healthy male individuals [15]. Conversely, it has been reported that dietary factors such as fiber, coffee, and green tea may have the potential to prevent the development and progression of NAFLD [14]. Earlier research on dietary interventions for NAFLD has suggested that following a Mediterranean-style diet may improve liver fat content even in the absence of weight loss [16]. The consumption of monounsaturated fatty acids found in olive oil, which is a prominent feature of the traditional Mediterranean diet [17], can contribute to a reduction in weight and fatty liver, as can the consumption of fish rich in omega-3 polyunsaturated fatty acids [18,19], which reduce the amount of neutral fat in the liver, and large amounts of vegetables, beans, fruits, and nuts [20]. The various dietary factors identified in the previous studies of NAFLD may also be effective in improving the fatty liver in non-obese NAFLD, where the range of weight loss required is limited. Therefore, the Mediterranean diet, which emphasizes olive oil, fish, and vegetable intake, with the avoidance of trans fatty acids, saturated fatty acids, cholesterol, and fructose, may be recommended for patients with non-obese NAFLD. However, abnormalities in eating habits limited to non-obese NAFLD are not clear, and no guidelines exist. Therefore, we hope to generate evidence that can be used for guidelines in the future.

It is widely accepted that exercise therapy is effective for NAFLD. In our laboratory, we found an improvement in hepatic steatosis in a group that performed moderate-to-high-intensity aerobic exercise for more than 250 min per week for 12 weeks [21]. For non-obese NAFLD, exercise interventions have been reported to achieve the remission of NAFLD with a modest weight loss of 5–10% [13]. However, the number of reports is small.

The etiology and optimal management strategy for non-obese NAFLD are still unclear and require further investigation, including dietary and physical activity interventions. The aim of this study, conducted via a questionnaire, was to investigate the associations between dietary habits, physical activity, and non-obese NAFLD.

## 2. Materials and Methods

### 2.1. Study Design and Subjects

This research had a cross-sectional design and was based on that used in a previous study [22]. The study participants were patients who were first referred to the outpatient clinic for lifestyle-related liver diseases at the University of Tsukuba Hospital after being found to be obese or to have hepatic dysfunction, dyslipidemia, glucose intolerance, or hypertension during medical or physical examination. Additional study participants were added in subsequent surveys. In this study, data from 455 patients who attended our clinic between January 2017 and December 2021 were analyzed. NAFLD was diagnosed via interview, ultrasonography, and ultrasound elastography. The following exclusion criteria were applied: alcohol intake > 30 g/day if male and 20 g/day if female [23]; other chronic liver disease; a diagnosis of NAFLD was not possible because no ultrasound examination was performed; refusal to participate; questionnaire not returned; or a BMI ≥ 25 (Figure 1).

### 2.2. Data Collection

The following clinical data used in this study were obtained from the medical records of the patients: anthropometric measurements (height, weight, BMI, body composition), muscle strength measurements (grip, knee extension strength), liver fat content and fibrosis (controlled attenuation parameters (CAPs), liver stiffness measurements (LSMs)), parameters related to liver function (albumin, aspartate transaminase (AST), alanine transaminase (ALT), alkaline phosphatase, gamma-glutamyl transferase, cholinesterase), glucose metabolism (fasting blood glucose, insulin, homeostasis model assessment-insulin resistance (HOMA-IR), HbA1c), lipid metabolism (total cholesterol, high-density lipoprotein cholesterol, low-density lipoprotein cholesterol, triglycerides), and blood pressure. The measurements were performed in the same way as described previously [24]. Obesity was defined as a BMI of ≥25, and sarcopenia-related indices, including the skeletal muscle index (limb skeletal muscle mass [kg]/height [m]^2^) [25], sarcopenia index (limb skeletal muscle mass [kg]/BMI) [26], and skeletal muscle mass to visceral fat area (SV) ratio [27], were calculated. Surrogate markers, including insulin resistance via HOMA-IR [28], the FIB-4 index [29], and the NAFLD fibrosis score (NFS) [30], were calculated using the biochemical data. The severity of liver fibrosis was assessed using the FAST score [31], which incorporates the liver stiffness measurement (LSM), controlled attenuation parameter (CAP) to assess liver fat content, and aspartate aminotransferase (AST) to assess liver inflammation.

The data on dietary habits and physical activity were obtained using a questionnaire survey administered to the study participants. The survey was performed in the same way as described in a previous report [22]. The food items in the questionnaire were selected based on the five food categories commonly included in dietary assessments, including staple foods, main dishes, side dishes, milk and dairy products, and fruits [32], as well as foods that have been reported to have an impact on the development and progression of NAFLD in previous studies, to ensure comprehensive coverage of dietary intake. Details on physical activity level (PAL) and exercise (Ex) have also been provided in previous reports [22]. The questionnaire included a section on oral medications. The relationship between drugs, the clinical data, and comorbidities was carefully examined with reference to the patient’s medical history.

### 2.3. Statistical Analysis

The statistical analysis was conducted with the use of the SPSS software package, version 26 (IBM Corp., Armonk, NY, USA). Categorical variables were expressed as frequency and percentage, and continuous variables were presented as mean ± standard deviation. All statistical tests were two-tailed, and statistical significance was set at the 5% level. The Mann–Whitney U test and the chi-squared test were used for between-group comparisons. Confounding by covariates measured between the non-obese NAFLD group and the non-NAFLD group was reduced through the use of the propensity score method [33]. A propensity score quantifying the probability of developing non-obese NAFLD was calculated using a logistic regression model. Covariates for inclusion in the propensity score model were selected based on their likelihood of being associated with non-obese NAFLD [3,11] and included age, sex, skeletal muscle mass, knee extension stretch, and PAL. A multivariate model was created with the propensity score as a predictor variable, and binomial logistic regression analysis was performed using a model with the presence or absence of non-obese NAFLD as the outcome and food intake frequency of ≥4 times/week as a factor. The results are reported as the odds ratio (OR) and corresponding 95% confidence interval (CI).

### 2.4. Ethics Approval and Consent to Participate

The study was approved by the Ethical Review Committee of the University of Tsukuba Hospital (approval number H26-181) and conducted in compliance with the principles of the 1975 Declaration of Helsinki. All study participants signed an informed consent document after receiving a detailed explanation of the purpose of the study.

## 3. Results

### 3.1. Cohort Formation and Patient Demographic and Clinical Characteristics

Out of the 455 patients who visited the outpatient clinic from January 2017 to December 2021, a total of 169 patients were included in the analysis. The patients were divided into a non-obese NAFLD group (*n* = 74) and a non-NAFLD group (*n* = 95) (Figure 1).

The patient characteristics between the two study groups are compared in Table 1. There was no significant between-group difference in age (*p* = 0.663). Patients in the non-obese NAFLD group were more likely to be male than those in the non-NAFLD group (47.3% vs. 28.4%, *p* = 0.012). Patients in the non-obese NAFLD group had a significantly higher BMI (22.6 ± 2.0 vs. 21.4 ± 2.1, *p* < 0.001), body fat mass (16.4 ± 3.9 kg vs. 14.1 ± 4.3 kg, *p* < 0.001), body fat percentage (28.0 ± 6.6% vs. 25.7 ± 7.8%, *p* = 0.021), waist-to-hip ratio (0.9 ± 0.1 vs. 0.8 ± 0.0, *p* < 0.001), and visceral fat cross-sectional area (74.6 ± 23.6 cm^2^ vs. 64.7 ± 26.9 cm^2^, *p* = 0.002). There was no significant between-group difference in the skeletal muscle index (*p* = 0.079) or sarcopenia index (*p* = 0.736); however, patients in the non-obese NAFLD group tended to have a lower SV ratio (*p* = 0.094). Patients in the non-obese NAFLD group had a significantly higher CAP (280.4 ± 39.5 dB/m vs. 190.5 ± 34.8 dB/m, *p* < 0.001). As with CAP, the non-obese NAFLD group had a significantly higher LSM (6.2 ± 3.3 kPa vs. 4.7 ± 3.4 kPa, *p* < 0.001). Serum AST (33.0 ± 17.1 U/L vs. 23.8 ± 8.8 U/L, *p* < 0.001), serum ALT (39.2 ± 30.9 U/L vs. 20.0 ± 12.8 U/L, *p* < 0.001), cholinesterase (350.1 ± 89.2 U/L vs. 319.4 ± 78.6 U/L, *p* = 0.001), and gamma-glutamyl transferase (50.8 ± 44.1 U/L vs. 32.3 ± 41.0 U/L, *p* < 0.001) were all significantly higher in the non-obese NAFLD group. All parameters related to glucose metabolism, including fasting blood glucose (118.3 ± 32.0 mg/dL vs. 101.9 ± 25.2 mg/dL, *p* < 0.001), HbA1c (6.2 ± 0.8% vs. 5.7 ± 0.7%, *p* < 0.001), insulin (13.3 ± 17.0 µU/mg vs. 6.0 ± 5.1 µU/mg, *p* < 0.001), and HOMA-IR (4.0 ± 6.0 vs. 1.5 ± 1.3, *p* < 0.001), were significantly higher in the non-obese NAFLD group. The serum high-density lipoprotein cholesterol concentrations were lower in the non-obese NAFLD group (51.2 ± 12.7 mg/dL vs. 69.7 ± 16.4 mg/dL, *p* < 0.001), while the serum low-density lipoprotein cholesterol concentrations were not significantly different between the groups (*p* = 0.493). The serum triglyceride concentrations were significantly higher in the non-obese NAFLD group (132.8 ± 81.2 mg/dL vs. 82.4 ± 43.1 mg/dL, *p* < 0.001). Patients in the non-obese NAFLD group had significantly higher levels of CRP (0.2 ± 0.3 ng/mL vs. 0.1 ± 0.2 ng/mL, *p* < 0.001) and ferritin (130.7 ± 129.4 ng/mL vs. 76.0 ± 57.6 ng/mL, *p* = 0.001). There was no significant between-group difference in terms of hyaluronic acid (*p* = 0.707), type IV collagen deposition (*p* = 0.755), or Mac-2 binding protein glycan isomer (*p* = 0.199). There was also no significant between-group difference in the FIB-4 index (*p* = 0.751) or NAFLD fibrosis score (*p* = 0.889); however, the FAST score was higher in the non-obese NAFLD group (0.24 ± 0.20 vs. 0.08 ± 0.09, *p* < 0.001). Patients in the non-obese NAFLD group were significantly more likely to have diabetes (37.5% vs. 12.1%, *p* < 0.001), hypertension (50.0% vs. 25.3%, *p* = 0.001), and dyslipidemia (70.0% vs. 33.8%, *p* < 0.001).

### 3.2. Questionnaire on Dietary Habits and Physical Activity

Responses to the questionnaire on dietary habits and physical activity were compared between the two study groups (Table 2). The non-obese NAFLD group had a lower frequency of consumption of fish and fish products (2.28 ± 0.61 times/week vs. 2.56 ± 0.7 times/week, *p* = 0.004) and olive oil or canola/rapeseed oil (2.4 ± 0.75 times/week vs. 2.66 ± 0.7 times/week, *p* = 0.031). However, the non-obese NAFLD group had a higher frequency of consumption of pastries and cake (2.09 ± 0.69 times/week vs. 1.79 ± 0.7 times/week, *p* = 0.003), snack foods and fried sweets (1.69 ± 0.68 times/week vs. 1.51 ± 0.7 times/week, *p* = 0.046), candy and caramels (1.55 ± 0.71 times/week vs. 1.30 ± 0.5 times/week, *p* = 0.011), salty foods (2.14 ± 0.73 times/week vs. 1.82 ± 0.8 times/week, *p* = 0.004), and pickles (2.47 ± 0.86 times/week vs. 1.98 ± 0.9 times/week, *p* < 0.001). 

The PAL was lower in the non-obese NAFLD group (1.6 ± 0.3 vs. 1.7 ± 0.5, *p* = 0.054), as was the Ex (8.0 ± 14.1 METs/hour/week vs. 11.5 ± 21.7 METs/hour/week, *p* = 0.485); the between-group difference was not significant.

### 3.3. Association between Non-Obese NAFLD and Food Intake Frequency

To determine whether food intake frequency was associated with non-obese NAFLD, logistic regression analysis was performed with the presence of NAFLD as the outcome, a food intake frequency of ≥4 times/week as the risk factor, and propensity variables calculated using the propensity score method as the covariates (Table 3). There was a significant association between non-obese NAFLD and the consumption of fish and fish products (OR 0.261, 95% CI 0.114–0.594; *p* = 0.001) and pickles (OR 2.880, 95% CI 1.264–6.558; *p* = 0.012) ≥4 times/week. The results did not change when the analysis was conducted using a model that included the prevalence of diabetes and dyslipidemia in the propensity score.

## 4. Discussion

As far as we are aware, this is the first study using a questionnaire to examine the association between dietary habits and physical activity in Japanese patients with non-obese NAFLD.

The results of the questionnaire revealed that patients with non-obese NAFLD consumed fish and fish products and olive oil or canola/rapeseed oil less frequently and consumed pastries and cake, snack foods and fried sweets, candy and caramels, salty foods, and pickles more frequently. Dietary risk factors for non-obese NAFLD have been reported to include an excessive intake of cholesterol and fructose [7]. In this study, there was no significant between-group difference in the frequency of intake of cholesterol-rich chicken eggs [34]; however, the frequency of intake of pastries and cake and of snack foods and fried sweets [34], which contain high levels of saturated fatty acids [35] that indirectly increase serum cholesterol, was higher in the patients with non-obese NAFLD. The background of excessive fructose intake has been attributed to a high consumption of soft drinks and juices containing corn syrup fructose [36]. In our study, the between-group difference in the frequency of soft drink consumption was not statistically significant, although there was a trend toward a higher frequency of consumption in patients with non-obese NAFLD. However, the frequency of intake of candy and caramels [34] with a higher amount of corn syrup fructose per gram was higher in patients with non-obese NAFLD. These findings are consistent with those of previous studies [7]. 

Our findings suggest that there are several previously unrecognized dietary risk factors for non-obese NAFLD. In this study, the consumption of fish and fish products and olive oil or canola/rapeseed oil was less frequent in the non-obese NAFLD group. Furthermore, the consumption of fish and fish products ≥ 4 times/week had a suppressive effect on the fatty liver. This finding suggests that patients with a non-obese NAFLD lack sufficient unsaturated fatty acids in their diet. A dietary pattern characterized by a high intake of olive oil and fish, known as the Mediterranean diet, has been reported to reduce liver fat [16]. Levy et al. [19] reported that ω3 polyunsaturated fatty acids reduce hepatic lipidosis, and the administration of ω3 polyunsaturated fatty acids has been shown to be effective in treating hepatic lipids in patients with NAFLD [37]. Olive oil and rapeseed/canola oil contain high levels of oleic acid [34], a monounsaturated fatty acid of the ω9 family that has the effect of reducing insulin resistance and lipidosis [38]. Olive oil has been demonstrated to reduce the accumulation of triglycerides in the liver in a rodent model of NAFLD [39], and once-daily rapeseed/canola oil consumption has been reported to be effective in the treatment of hepatic adiposity [40]. The Mediterranean diet may have a favorable effect on the fatty liver because of the unsaturated fatty acids contained in fish and olive oil. 

In a previous study by Zhao et al. [41], it was found that a diet high in salt can have an impact on hypertriglyceridemia, oxidative stress, and inflammation, whereby all of which are known pathological factors in the development of NAFLD, and a higher prevalence of NAFLD has been reported in young and middle-aged adults who consume a high-sodium diet, according to a previous study [42]. In our study, patients with non-obese NAFLD had a higher frequency of consumption of salty foods and of pickles, which contain a high amount of salt. Furthermore, the consumption of pickles ≥ 4 times/week promoted the formation of the fatty liver. This finding suggests that a high-salt diet may contribute to the formation of NAFLD even in non-obese individuals. 

Body composition analysis showed that an abnormal composition such as an increase in the internal fat cross-sectional area and a decreased tendency toward an SV ratio of sarcopenia body mass index was observed in those with non-obese NAFLD. It has been thought that visceral fat and muscle balance abnormalities are associated with the pathophysiology of non-obese NAFLD [11]. Our laboratory has reported that exercise intervention in patients with NAFLD has little effect on weight loss, but maintains lean body (skeletal muscle) mass and increases large muscle strength [43], suggesting that exercise improves the pathophysiology of NAFLD independently of weight loss. Given that the responses to our questionnaire showed that patients with non-obese NAFLD tended to have low PAL and Ex (Table 2), we believe that in addition to their daily PAL they should perform further voluntary exercise to maintain a good balance between visceral fat and muscle mass.

This study had several limitations. First, the Food Frequency Questionnaire was used as the dietary survey method. Dietary survey methods, regardless of which method is used, are reportedly subject to errors as a result of diurnal variation [44] and both under-reporting and over-reporting by the respondents [45]. Second, because of the small number of patients in the non-obese NAFLD group, the propensity score method was used to eliminate the possibility of over-fitting when performing multivariate analysis; however, the study was retrospective in nature and the ability to control for differences was limited to variables for which data were available. Third, it should be noted that this was not designed as a prospective cohort trial and was conducted in a single hospital, making it likely to lead to spurious causality and bias. Therefore, in the near future, we are planning a prospective cohort study based on the results from this study.

## 5. Conclusions

Our questionnaire-based study revealed that a lower frequency of fish and fish product consumption, as well as a higher frequency of pickled food consumption, were associated with an increased risk of non-obese NAFLD. These findings suggest that dietary habits and physical activity should be taken into account in the routine clinical management of Japanese patients with non-obese NAFLD.

## Figures and Tables

**Figure 1 nutrients-15-02764-f001:**
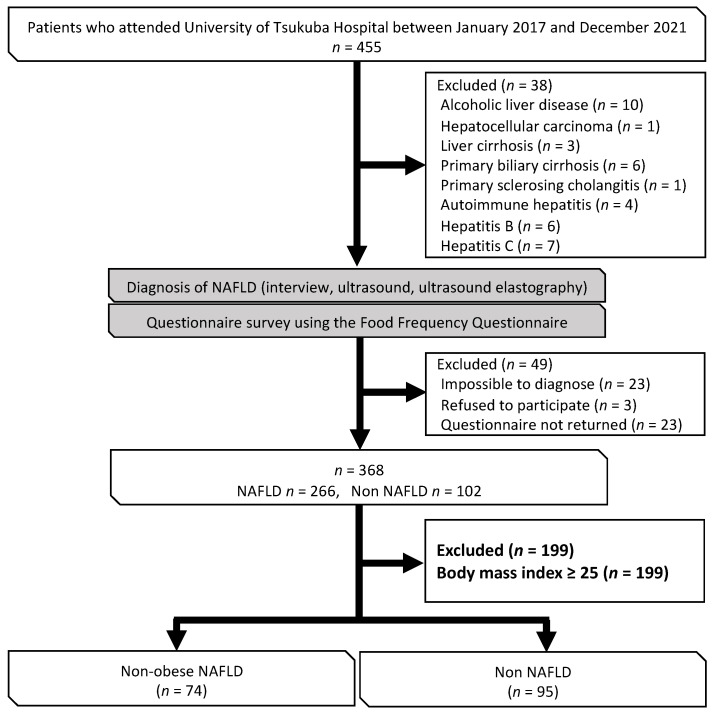
Flow chart showing the process used to select the study participants.

**Table 1 nutrients-15-02764-t001:** Patient characteristics.

	Non-Obese NAFLD			Non-NAFLD			
	Mean		SD	Mean		SD	*p*-Value
Age, years	61.7	±	14.2	58.7	±	18.0	0.663
Male sex, *n* (%)	35 (47.3)			27 (28.4)			0.012
Body mass index	22.6	±	2.0	21.4	±	2.1	<0.001
Fat, kg	16.4	±	3.9	14.1	±	4.3	<0.001
Percent body fat, %	28.0	±	6.6	25.7	±	7.8	0.021
Waist-to-hip ratio	0.9	±	0.1	0.8	±	0.0	<0.001
Skeletal muscle mass, Kg	23.0	±	5.0	21.9	±	5.1	0.121
Arm muscle, kg	2.1	±	0.6	1.9	±	0.6	0.018
Trunk muscle, kg	18.8	±	3.8	17.6	±	3.9	0.022
Leg muscle, kg	6.4	±	1.5	6.2	±	1.5	0.230
Total body water, L	31.1	±	6.0	29.7	±	5.8	0.111
Soft lean mass, kg	39.9	±	7.8	38.4	±	7.8	0.159
Fat free mass, kg	42.2	±	8.1	40.1	±	7.9	0.081
Visceral fat cross-sectional area, cm^2^	74.6	±	23.6	64.7	±	26.9	0.002
SMI, kg/m^2^	6.5	±	0.9	6.3	±	0.9	0.079
SI, kg/(kg/m^2^)	0.8	±	0.2	0.8	±	0.2	0.736
SV ratio, kg/cm^2^	339.4	±	129.4	408.7	±	246.0	0.094
Hand grip, kgf	28.9	±	7.9	27.0	±	8.3	0.072
Knee extension stretch, kgf	36.0	±	10.6	34.8	±	12.4	0.201
LSM, kPa	6.2	±	3.3	4.7	±	3.4	<0.001
CAP, dB/m	280.4	±	39.5	190.5	±	34.8	<0.001
TP, g/dL	7.4	±	0.5	7.2	±	0.5	0.032
Albumin, g/dL	4.4	±	0.3	4.3	±	0.3	0.120
AST, U/L	33.0	±	17.1	23.8	±	8.8	<0.001
ALT, U/L	39.2	±	30.9	20.0	±	12.8	<0.001
LD, U/L	174.6	±	33.4	183.2	±	40.5	0.309
ALP, U/L	207.8	±	99.8	189.4	±	88.9	0.211
CHE, U/L	350.1	±	89.2	319.4	±	78.6	0.001
γ-GT, U/L	50.8	±	44.1	32.3	±	41.0	<0.001
Creatinine, mg/dL	0.7	±	0.2	0.7	±	0.2	0.178
Fasting glucose, mg/dL	118.3	±	32.0	101.9	±	25.2	<0.001
HbA1c, %	6.2	±	0.8	5.7	±	0.7	<0.001
Insulin	13.3	±	17.0	6.0	±	5.1	<0.001
HOMA-IR	4.0	±	6.0	1.5	±	1.3	<0.001
CHO, mg/dL	199.8	±	44.3	210.8	±	37.0	0.094
HDL-C, mg/dL	51.2	±	12.7	69.7	±	16.4	<0.001
LDL-C, mg/dL	119.9	±	35.5	116.9	±	28.5	0.493
Triglycerides, mg/dL	132.8	±	81.2	82.4	±	43.1	<0.001
Platelets, mg/dL	226.9	±	62.9	221.0	±	56.4	0.960
CRP, ng/mL	0.2	±	0.3	0.1	±	0.2	<0.001
Ferritin, ng/mL	130.7	±	129.4	76.0	±	57.6	0.001
Hyaluronic acid, ng/mL	51.6	±	54.4	48.2	±	48.5	0.707
Type IV collagen, ng/mL	118.3	±	36.2	116.1	±	42.9	0.755
M2BPGi, C.O.I.	0.8	±	0.5	0.7	±	0.5	0.199
FAST score	0.24	±	0.20	0.08	±	0.09	<0.001
FIB-4 index	1.76	±	1.12	1.96	±	3.03	0.751
NAFLD fibrosis score	−1.67	±	1.78	−1.64	±	1.61	0.889
Systolic BP, mmHg	133.1	±	17.2	123.9	±	14.9	<0.001
Diastolic BP, mmHg	78.9	±	12.4	73.8	±	10.3	0.002
Diabetes mellitus, *n* (%)	27 (37.5)			11 (12.1)			<0.001
Hypertension, *n* (%)	37 (50.0)			24 (25.3)			0.001
Dyslipidemia, *n* (%)	49 (70.0)			27 (33.8)			<0.001

Categorical data were reported as frequencies and percentages, while continuous data were presented as mean and standard deviation. The appropriate statistical tests, such as the Mann–Whitney U test and chi-squared test, were conducted for group comparisons. The diagnosis of diabetes mellitus was based on the criteria established by the Japanese Diabetes Society, which include a fasting blood glucose level of ≥126 mg/dL, an oral glucose tolerance test result of ≥200 mg/dL at 2 h, a random blood glucose level of ≥200 mg/dL, or an HbA1c of ≥6.5%, as confirmed by a physician. Hypertension was determined based on either the use of antihypertensive medication or meeting the criteria of systolic blood pressure ≥ 140 mmHg and diastolic blood pressure ≥ 90 mmHg, as per the guidelines set by the Japanese Society of Hypertension. Dyslipidemia was determined based on either a physician diagnosis and treatment with lipid-lowering drugs or meeting the following criteria according to the Japanese Society for Arteriosclerosis: serum LDL-C ≥ 140 mg/dL, triglycerides ≥ 150 mg/dL, and HDL-C < 40 mg/dL. ALP: alkaline phosphatase, ALT: alanine aminotransferase, AST: aspartate transaminase, BP: blood pressure, CAP: controlled attenuation parameter, CHE: cholinesterase, CHO: cholesterol, CRP: C-reactive protein, FAST: FibroScan-AST, γ-GT: gamma-glutamyl transferase, HbA1c: glycated hemoglobin, HDL-C: high-density lipoprotein cholesterol, HOMA-IR: homeostasis model assessment–insulin resistance, LD: lactate dehydrogenase, LDL-C: low-density lipoprotein cholesterol, LSM: liver stiffness measurement, M2BPGi: Mac-2 binding protein glycosylation isomer, NAFLD: non-alcoholic fatty liver disease, SD: standard deviation, SI: sarcopenic index, SMI: skeletal muscle mass index, SV ratio: skeletal muscle mass to visceral fat area ratio, TP: total protein.

**Table 2 nutrients-15-02764-t002:** Comparison of responses to the questionnaire on dietary habits and physical activity.

		Non-Obese NAFLD			Non-NAFLD			
		Mean		SD	Mean		SD	*p*-Value
PAL		1.60	±	0.3	1.7	±	0.5	0.054
Ex	METs/hours/week	8.00	±	14.1	11.5	±	21.7	0.485
Potatoes, pumpkin, and lotus root	times/week	2.15	±	0.46	2.19	±	0.7	0.982
Fruit	times/week	2.59	±	0.91	2.82	±	0.9	0.089
Meat and meat products	times/week	2.82	±	0.69	2.71	±	0.7	0.362
Fish and fish products	times/week	2.28	±	0.61	2.56	±	0.7	0.004
Seafood	times/week	2.05	±	0.58	2.17	±	0.7	0.269
Eggs	times/week	2.77	±	0.73	2.82	±	0.8	0.647
Soybeans and soybean products	times/week	2.96	±	0.78	2.97	±	0.8	0.749
Milk and milk products	times/week	3.11	±	0.85	3.17	±	0.9	0.647
Seaweed	times/week	2.35	±	0.67	2.42	±	0.8	0.650
Small fish	times/week	1.92	±	0.64	1.93	±	0.8	0.687
Green and yellow vegetables	times/week	3.23	±	0.73	3.37	±	0.8	0.156
Light-colored vegetables	times/week	2.96	±	0.80	3.12	±	0.8	0.201
Oligosaccharide	times/week	1.53	±	0.69	1.57	±	0.7	0.818
Jam and honey	times/week	2.22	±	0.90	2.38	±	0.9	0.226
Simmered food	times/week	2.32	±	0.64	2.42	±	0.6	0.338
Vinegared and dressed vegetables	times/week	2.00	±	0.72	2.03	±	0.8	0.780
Japanese sweets	times/week	1.89	±	0.64	1.80	±	0.7	0.255
Pastries and cake	times/week	2.09	±	0.69	1.79	±	0.7	0.003
Snack foods and fried sweets	times/week	1.69	±	0.68	1.51	±	0.7	0.046
Rice crackers and cookies	times/week	2.05	±	0.59	1.90	±	0.8	0.062
Ice cream	times/week	1.77	±	0.73	1.60	±	0.7	0.111
Chocolate	times/week	2.00	±	0.79	1.82	±	0.7	0.152
Candy and caramels	times/week	1.55	±	0.71	1.30	±	0.5	0.011
Jelly and pudding	times/week	1.55	±	0.64	1.44	±	0.5	0.346
Soft drink	times/week	2.04	±	1.05	1.86	±	0.9	0.337
Fried food	times/week	2.18	±	0.56	2.04	±	0.5	0.100
Stir-fries	times/week	2.38	±	0.61	2.45	±	0.6	0.513
Mayonnaise and dressing	times/week	2.41	±	0.74	2.39	±	0.9	0.763
Margarine and fat spread	times/week	1.73	±	0.82	1.54	±	0.8	0.079
Butter, lard, and beef tallow	times/week	1.68	±	0.70	1.65	±	0.8	0.595
Peanuts and almonds	times/week	1.86	±	0.80	1.98	±	0.9	0.479
Sesame	times/week	2.00	±	0.64	2.00	±	0.7	0.994
Olive oil and canola/rapeseed oil	times/week	2.40	±	0.75	2.66	±	0.7	0.031
Safflower oil, cottonseed oil, and soybean oil	times/week	1.46	±	0.76	1.47	±	0.8	0.961
Perilla oil, egoma oil, and flaxseed oil	times/week	1.41	±	0.66	1.50	±	0.9	0.996
Salty foods	times/week	2.14	±	0.73	1.82	±	0.8	0.004
Pickles	times/week	2.47	±	0.86	1.98	±	0.9	<0.001
Tabletop soy sauce and sauces	times/week	2.71	±	0.77	2.61	±	0.9	0.475
Miso soup	times/week	2.65	±	0.90	2.76	±	0.9	0.383
Soup other than miso soup	times/week	1.82	±	0.58	1.70	±	0.7	0.125
Noodles	times/week	2.05	±	0.55	1.98	±	0.6	0.383

The mean and standard deviation were used to describe the values, and the between-group comparisons were performed using the Mann–Whitney U test. NAFLD: non-alcoholic fatty liver disease, PAL: physical activity level, Ex: exercise.

**Table 3 nutrients-15-02764-t003:** Logistic regression analysis of the association between dietary intake frequency and non-obese NAFLD.

	NAFLD			
	OR	95% CI		*p*-Value
		Lower	Upper	
Propensity score	218.518	14.177	3368.249	<0.001
Fish and fish products ≥ 4 times/week	0.261	0.114	0.594	0.001
Pastries and cake ≥ 4 times/week	1.801	0.588	5.515	0.303
Candy and caramels ≥ 4 times/week	3.097	0.302	31.704	0.341
Pickles ≥ 4 times/week	2.880	1.264	6.558	0.012

Outcome: NAFLD. Factors: fish and fish products, pastries and cake, candy and caramels, and pickles ≥ 4 times/week. Adjusted for propensity score that included age, sex, skeletal muscle mass, knee extension stretch, and PAL. NAFLD: non-alcoholic fatty liver disease, OR: odds ratio, CI: confidence interval.

## Data Availability

The datasets analyzed in this study are available from the corresponding author upon reasonable request. Further inquiries can be directed to the corresponding author.

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
