# Peer review of "Survey of Dietary Habits and Physical Activity in Japanese Patients with Non-Obese Non-Alcoholic Fatty Liver Disease"

_nutrients, 2023, doi:10.3390/nu15122764_

Round 1
Reviewer 1 Report
The study used single-center retrospective cohort design to investigate the association between non-obese NAFLD and dietary habits and physical activity levels. This research might contribute to the development for the management of non-obese NAFLD. However, due to the risk factors of lipid metabolism having be well-known, fewer highlights are found in the study. There are several questions needed to be elucidated also.
The study looks more like a case-control design rather than a cohort study. The time interval between the questionnaire/clinical record collection and the outcomes should be stated. Moreover, there may be an incomparability in observation time when a large difference of interval was existed among patients. How did the authors deal with this?
Exclusion criteria included alcohol 80 intake > 30 g/day if male and 20 g/day if female. Which guideline or reference was the criteria according to? The authors applied a food intake frequency of ≥ 4 times/week as the risk factor of NAFLD. A reference is necessary as well.
There are so many characteristics/factors in the regression that may have increased the uncertainty for the model. Despite using a propensity score method, a big bias cannot be avoided due to a relatively small sample size. There may be some collinear correlations among those characteristics. The model is suggested to be modified with fewer characteristics by using a selection of important contribution to model.
In results, section 3.1 and 3.2 can be combined to present together, because they all described the results of Table 1.
Figure 1 does not exist in the article.
0.000 of p value in Tables should be replaced with <0.001.
Biomarkers or clinical data can be altered due to medicine (lipid-lowering drugs, etc.) or treatment for patients. How can the study clarify the factors independently and irrespectively to the effects of medicine?
An IRB approval should be added.
Reviewer 2 Report
In my opinion, the manuscript is generally well written and the results are interesting. Nevertheless, I have the following comments:
- Introduction - The potential association of physical activity with NAFLD should be clarified
- Figure 1 referred to in line 140 is missing
- Table 1 - In my opinion, the first 3 items in the table (diabetes, hypertension, dyslipidemia) should be at the end of the table as they do not indicate the basic parameters describing the study group. It might even be better to include these parameters in a separate table.
- Section 3.2 lacks reference to Table 1.
Reviewer 3 Report
The topic of the article on eating habits and physical activity in Japanese patients with non-alcoholic fatty liver disease is quite interesting. In the introduction, the authors write (line 57) that “…the optimal diet for patients with non-obese NAFLD is not known…”, which I cannot agree with. I suggest you read the publication “Dietary Recommendations for the Management of Non-alcoholic Fatty Liver Disease (NAFLD): A Nutritional Geometry Perspective” (Semin Liver Dis 2022; 42(04): 434-445). The introduction lacks a description of the optimal diet in NAFLD - which products are recommended and which should be excluded.
In addition:
1. line 35 - remove keyword 1 and numbers 2,3,4,5.
2. Expand the NAFLD abbreviation on first use.
3. NAFLD is common in people who have metabolic syndrome, diabetes mellitus, hypothyroidism, or genetics. This topic should be developed in the introduction and discussion.
4. Did the people included in the study use any medications (e.g. Diabetics?). If so, how could they modify the determined parameters? Please discuss it.
Round 2
Reviewer 1 Report
1. Due to a cross-sectional design, the study has a major limitation to explore the cause-effect relationship between the risk factors and NAFLD. The findings have been doubted and may contribute less knowledge to current science. The authors are required to state how they had diminished the uncertainties as possible as they can.
2. Though there is no significant difference of oral medications between the non-obese NAFLD group and the non-NAFLD group, the differences of percentages for the complications such as diabetes and dyslipidemia are large obviously. I consider that the authors should regenerate a model which includes the factors.
3. The Fig.1 is suggested to move to Method section. The number of inclusion may have some errors. The number of 368 should be diminished to 167 (not 169) after excluding 199.
4. The guidelines of alcohol and food intake the study used should be presented in the article.
